# Effects of Artificial Light at Night (ALAN) on European Hedgehog Activity at Supplementary Feeding Stations

**DOI:** 10.3390/ani10050768

**Published:** 2020-04-28

**Authors:** Domhnall Finch, Bethany R. Smith, Charlotte Marshall, Frazer G. Coomber, Laura M. Kubasiewicz, Max Anderson, Patrick G. R. Wright, Fiona Mathews

**Affiliations:** 1School of Life Sciences, University of Sussex, Falmer BN1 9QG, UK; d.finch@sussex.ac.uk (D.F.); science@themammalsociety.org (F.G.C.); Max.Anderson@sussex.ac.uk (M.A.); P.G.R.Wright@sussex.ac.uk (P.G.R.W.); 2Mammal Society, London E9 6EJ, UK; bethanys935@gmail.com (B.R.S.); charlotte.marshall401@gmail.com (C.M.); laurakuba@googlemail.com (L.M.K.)

**Keywords:** activity pattern, ALAN, camera trap, citizen science, fragmentation, hedgehogs, *Erinaceus europaeus*, light pollution, lightscape, urbanisation

## Abstract

**Simple Summary:**

Owing to the rapid expansion of urbanisation, light pollution has increased dramatically in the natural environment causing significant negative effects on species fitness, abundance, foraging and roosting behaviours. However, very little research has examined the impacts of artificial light at night (ALAN) on mammal species other than bats. Using a large-scale citizen science project, we examined the potential impact of ALAN on European hedgehogs (*Erinaceus europaeus*) at supplementary feeding stations. Our results show that there were no significant effects of ALAN on the presence, feeding activity or activity patterns of hedgehogs throughout the experiment, although some variations in individual hedgehogs were observed. This suggests that while there was no significant impact of ALAN found at supplementary feeding stations, there could be other costs associated with lighting, e.g., reproductive success, territory maintenance and natural prey availability, which need to be considered.

**Abstract:**

Artificial light at night (ALAN) can have negative consequences for a wide range of taxa. However, the effects on nocturnal mammals other than bats are poorly understood. A citizen science camera trapping experiment was therefore used to assess the effect of ALAN on the activity of European hedgehogs (*Erinaceus europaeus*) at supplementary feeding stations in UK gardens. A crossover design was implemented at 33 gardens with two treatments—artificial light and darkness—each of which lasted for one week. The order of treatment depended on the existing lighting regime at the feeding station: dark treatments were applied first at dark feeding stations, whereas light treatments were used first where the station was already illuminated. Although temporal changes in activity patterns in response to the treatments were noted in some individuals, the direction of the effects was not consistent. Similarly, there was no overall impact of ALAN on the presence or feeding activities of hedgehogs in gardens where supplementary feeding stations were present. These findings are somewhat reassuring insofar as they demonstrate no net negative effect on a species thought to be in decline, in scenarios where the animals are already habituated to supplementary feeding. However, further research is needed to examine long-term effects and the effects of lighting on hedgehog prey, reproductive success and predation risk.

## 1. Introduction

Urbanisation is increasing rapidly across the globe [1], with important impacts for biodiversity and ecosystem functioning [2,3]. One of the major environmental changes that accompanies urbanisation is an increase in the amount and intensity of artificial light at night (ALAN), and this has become a significant conservation concern [4]. Reported impacts of ALAN include the disruption of predator–prey interactions [5], seed dispersal [6], foraging [7], and migratory behaviour [8]. 

Lunar cycles have been shown to elicit a variety of behavioural and physiological responses in nocturnal mammals [9,10,11]. Given that the illuminance from streetlights is usually more than an order of magnitude greater than that from a full moon, it is likely that nocturnal mammals are susceptible to the effects of ALAN. Most research to date has focused on bats, partly because of their high legislative protection in some parts of the world (e.g., Europe) [12,13]. However, there has been limited research on other nocturnal mammals. 

The European hedgehog (*Erinaceus europaeus*) is a relatively small nocturnal insectivore that has a wide distribution across Europe. However, populations are thought to be in decline. In Britain, for example, populations have reduced from approximately 1.5 million individuals in 1995 to 522,000 in 2016 [14], and concerns have been raised in other countries [15,16,17,18]. The mechanisms underlying these declines remain unclear, though a wide variety of pressures—including predation by and competition with badgers [19,20,21], road collisions [22,23], and agricultural intensification [24]—have been suggested. 

Hedgehogs make extensive use of parks and gardens, particularly in peri-urban and suburban areas, where they can obtain shelter and food (including from supplementary feeding by householders, which is common in Britain) [15,25]. However, while these areas are widely viewed as strongholds for the species, close interactions with humans also present challenges. For example, roads act as an important barrier to movement, resulting in isolated populations [26] and direct hedgehog mortalities [27]. The amount and intensity of ALAN associated with built environments is increasing [28,29], and the implications of this change for hedgehogs and their prey are relatively unknown. This study uses a citizen science approach to assess whether the presence of ALAN in gardens is linked to (1) the amount of time hedgehogs are present at supplementary feeding stations; (2) the amount of time hedgehogs spend feeding at supplementary feeding stations and (3) hedgehog activity patterns in those gardens.

## 2. Materials and Methods

### 2.1. Survey

Initial questionnaires were sent out to any member of the public interested in taking part in this experiment, asking for details about their gardens and the environmental features surrounding them, e.g., streetlights. Any participant who had features likely to interfere with the experiment was excluded. Camera traps (Bushnell Trophy CAM HD Max/Bushnell NatureView CAM HD Max; Bushnell Corporation, Overland Park, Kansas, United States) were then sent to 33 volunteer citizen scientists throughout England and Wales who had indicated that hedgehogs visited feeding stations in their gardens (Appendix A). Each volunteer received one camera trap that they deployed in their garden (hereafter referred to as site) facing pre-existing feeding stations. To prevent disturbance to the hedgehogs beyond the experimental treatment, volunteers were asked to continue to supply the same fish-free food they usually provided. The camera traps were placed 2–4 m away from feeding stations and were elevated approximately 0.5 m from the ground to avoid interference with the artificial light source used during the experiment. The cameras were set to record 60 s of video, with an interval of at least 60 s between videos. 

At each site, two lighting treatments were used for one week each—‘dark’ (no artificial lighting illuminating the feeding station) and ‘light’ (constant artificial light source illuminating the feeding station during the night)—and each treatment lasted for seven nights. The two treatments were deployed sequentially, beginning with the treatment that was already present at the site (i.e., sites that already had illuminated feeding stations began with the ‘light’ treatment, whereas the opposite was true for ‘dark’ feeding stations). The two treatments were applied in consecutive weeks using a paired design, and the experiments took place between 15 July and 21 August 2017. For sites that were initially ‘dark’, an LED floodlight of approximately 1000 lumens (Powerline rechargeable flood lights, 10 W 115 Lm/W) was supplied if the participants did not have a suitable bright exterior light available (e.g., a patio light). The lights were placed so that they constantly illuminated the feeding station throughout the entirety of the night.

The times at which hedgehogs were present and feeding from the supplementary feeding stations were then recorded from the camera trap videos. Supporting data for this study have been deposited on Figshare digital repository (10.6084/m9.figshare.11872113). To reduce the impact of pseudo-replication caused by taking multiple measurements of the same individual (for example, if one animal was stationary in front of the camera for 5 min), a presence index was created by classifying 10 min recording blocks as either being positive or negative for hedgehogs, and a similar index for feeding was created depending on whether they fed during this interval. To ensure ease of recording for volunteers, recording blocks were split by clock time (e.g., 21:00–21:10, and then 21:10–21:20). So if a hedgehog was present over both blocks, this counted as two positive recording blocks, even if the hedgehog was present for less than 10 min. It is highly likely that multiple individuals would have visited the same feeder within each night, but it was not possible to recognise individuals in this project. Therefore, for the purposes of analysis, each 10 min recording block was considered to be a replicate, and the nightly count of hedgehog-positive (the ‘presence index’) and feeding-positive (the ‘feeding index’) recording blocks were used as outcome variables. 

The percentage of urban cover (combined urban and suburban) within a circular buffer centred on each site was calculated from the Land Cover Map 2007 [30] using ArcMAP 10.5 [31]. The area of the buffer was set at 9.7 ha, which has been reported as the mean home range of hedgehogs in England in regions where badgers are present [32].

### 2.2. Statistical Analysis

Statistical analyses were undertaken using R v.3.5.3 [33]. To assess the relationships between light exposure and the indices of hedgehog presence/feeding activity, Generalised Linear Mixed Models with a negative binomial distribution were fitted to the count data using the ‘lme4’ package [34]. Using a paired design, the full model had the following predictor variables: treatment, treatment order, the interaction between treatment and treatment order, percentage of urbanisation, the type of light used (supplied by study participant or the researchers) and an interaction between percentage of urbanisation and treatment type. Site was included as a random effect (accounting for some variation between sites), and an offset for night-length was also included to account for the increase in the time available for nocturnal activity that occurred across the course of the study. The percentage of urban cover was included to account for potential differences in hedgehog behaviour between more urban and more rural areas. Treatment, the random effect (site), and the offset were included in all models, and the most parsimonious model was identified using stepwise deletion of the fixed effects and inspection of the AIC values. 

Activity patterns for 31 sites were calculated following a nonparametric kernel density approach [35] using the package ‘activity’ [36,37]. Sites 1 and 14 were removed from this analysis because they had sample sizes of less than 10 [38]. The times of camera detections were converted to radians and were used to build circular kernel Probability Density Functions (PDF), which approximate the underlying activity pattern of the animals [35]. These PDFs were generated for the ‘dark’ and ‘light’ treatment at each site in a pairwise manner.

To investigate whether fitted activity patterns differed according to the lighting treatment, the coefficient of overlap (Δ)—a continuous variable that ranges from 0 (no overlap) to 1 (complete overlap) [35]—was calculated for each site. Then, a randomisation test with 1000 bootstrap iterations was run to generate a null distribution of randomised overlap values, followed by a Wald test to estimate the probability that the observed overlap arose by chance [38].

## 3. Results

Throughout the study period, night length ranged from 459 to 586 minutes (mean: 509 minutes), equating to an average of 51 possible 10 min recording blocks per site, per night. Of the total 22,615 recording blocks surveyed, 3470 (15%) contained hedgehogs and 19,145 did not. Of the 3470 hedgehog-positive recording blocks, 1724 occurred in ‘dark’ and 1746 in ‘light’ treatments, respectively. The mean was 7.8 (SD: 5.9) hedgehog-positive blocks per night across all sites, with a maximum number of hedgehog-positive recording blocks per night under dark treatments of 27 (mean: 7.7; SD: 5.5) and 36 blocks (mean: 7.9; SD: 6.2) for light treatments. The maximum number of animals seen within a single recording block was five. There was no evidence of a link between any of the fixed factors and their interactions (treatment order, percentage of urbanisation, type of light used) and hedgehog presence at the supplementary feeding station (*p* > 0.1 in each case of stepwise removal). The odds ratio for just treatment in the final model was 1.00 (95% Confidence Interval: 0.92–1.09, *p* = 0.945; Figure 1). 

Feeding activity was recorded in 2673 of the 3470 10 min recording blocks where hedgehogs were present, with a mean of 6.0 blocks per night across all sites (SD: 4.8). Hedgehogs were recorded feeding in 78% and 76% of the hedgehog-positive recording blocks in the ‘dark’ and ‘light’ treatments, respectively. The nightly number of feeding-positive blocks was similar under the different treatments (mean: 6.1; SD: 4.8 for ‘dark’ and mean: 6.0; SD: 4.8 for ‘light’ treatments). There was no evidence of a link between any of the fixed factors and the hedgehog feeding activity index (*p* > 0.2 in each case in stepwise removal). In the final model, the odds ratio for treatment was 0.99 (95% CI: 0.91–1.10, *p* = 0.992).

The circadian patterns of hedgehog activity did not vary between treatments at 18 of 31 sites (58%), with high levels of overlap being recorded (mean Δ: 0.93; SD: 0.06 (see Figure 2 for example)). However, at the other 13 sites, activity patterns differed significantly between the ‘light’ and ‘dark’ treatments (mean Δ: 0.72; SD: 0.09); see supplementary material for individual test statistics (Appendix A) and activity pattern plots (Appendix A). Furthermore, the observed differences in the responses of hedgehog activity at these sites to light treatments were variable, with no consistent directional changes in peak activity times or duration of activity. The peak activity occurred at a similar time at five sites (38.5%), was later in dark compared with light at four sites and was earlier in dark compared with light at four sites. Similarly, the duration of activity in light compared with dark was similar at nine sites (69.2%), was longer at two sites, and was shorter at two other sites. A bimodal pattern of activity was observed during 32% sites during dark treatment and 45% of sites during light treatments (Appendix A).

## 4. Discussion 

The disruptive effects of ALAN on the behaviour and biological processes of a wide range of taxa are well known [39,40,41,42,43]. However, our results indicate that there is no consistent overall effect of lighting on indices of hedgehog activity and feeding at supplementary feeding stations, or on the timing of these behaviours. These findings support those of De Molenaar et al. [44], who found no significant effect of ALAN on hedgehog crossing behaviour at roads. Although it is not possible to rule out the possibility of type-II errors (false negatives) categorically, the use of a controlled crossover trial should have minimised the effect of confounding factors, such as the structure of the garden and local food availability. Our results do, however, offer some support to previous findings that hedgehogs display bimodal feeding activity [45,46]. 

Although most studies report negative behavioural consequences of ALAN in mammals, some species are able to exploit the foraging opportunities created by lighting, such as the accumulation of insects around streetlights [12,47,48,49,50,51,52]. In contrast, an experimental study on the beech mouse (*Peromyscus polionotus leucocephalus*) showed that supplementary feeding stations were less likely to be visited, and had less food removed, when they were exposed to ALAN [53]. These results were attributed to a greater predation risk at lit feeding stations, although the degree of this risk will depend on the degree of light-tolerance in the predators. The most common prey items of hedgehogs are ground beetles (*Carabidae*) [54,55], which have been shown to be more abundant in artificially lit areas compared to dark [56]; though other taxa that feature in hedgehog diet, such as woodlice (*Onisicidea*), are light averse. The main predator of hedgehogs in the UK is the European badger (*Meles meles*) [57]. The activity and distribution of badgers has been shown to influence that of hedgehogs in urban and suburban environments [58], with hedgehogs tending to be more active in smaller gardens, which are less likely to be visited by badgers [59]. However, supplementary feeding stations provide very energy-dense resources compared with natural foods, and the drive to obtain these resources may outweigh the risk of predation or light-avoidance behaviours.

No consistent effects of ALAN were reported in this study, and this may in part be because of individual differences, i.e., sex and age. For example, male hedgehogs are bolder [59] and have larger home range sizes than females [60]. Changes in individual responses may explain the marked variability between sites (as illustrated in Appendix A). Differences between the sexes in response to ALAN are reported in great tits (*Parus major*), with females spending more time awake under ALAN conditions than males [61]. In this project, it was not possible to identify individuals or classify their age and sex. Therefore, it would be useful to conduct a similar study, with individuals of known age and sex.

## 5. Conclusions

In conclusion, this study revealed that the use of artificial light at night had no overall effect on the feeding and general activity of hedgehogs at supplementary feeding stations. There was also no evidence for any overall impact on the periodicity of activity: whilst some individuals delayed their activity when exposed to light, the reverse was true for other individuals. Despite the lack of any difference in hedgehog activity between lit and unlit treatments, there may be costs for reproductive success, territory maintenance, predation rate, and natural prey availability. Future research should focus on these areas.

## Figures and Tables

**Figure 1 animals-10-00768-f001:**
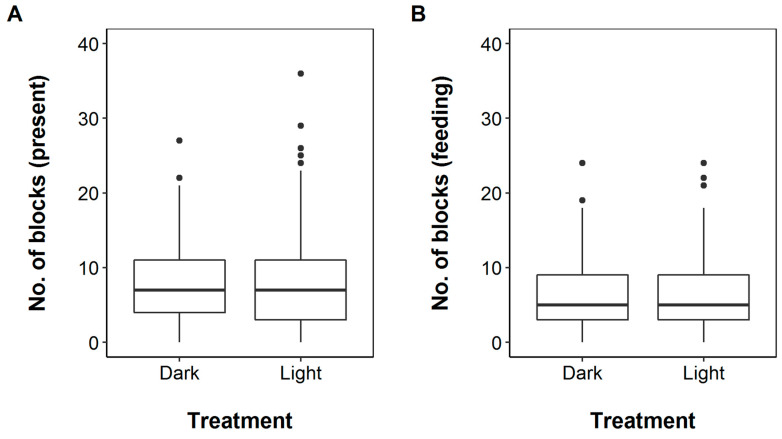
Box plot illustrating the effects of artificial light treatments at supplementary feeding stations on hedgehog (**A**) presence and (**B**) feeding records. ‘No. of blocks’ represents the number of 10 min periods per site, per night, in which hedgehogs were present and/or feeding. The total possible number of blocks, based on night lengths, ranged from 46 to 59, with an average of 51. Plots are based on the raw data.

**Figure 2 animals-10-00768-f002:**
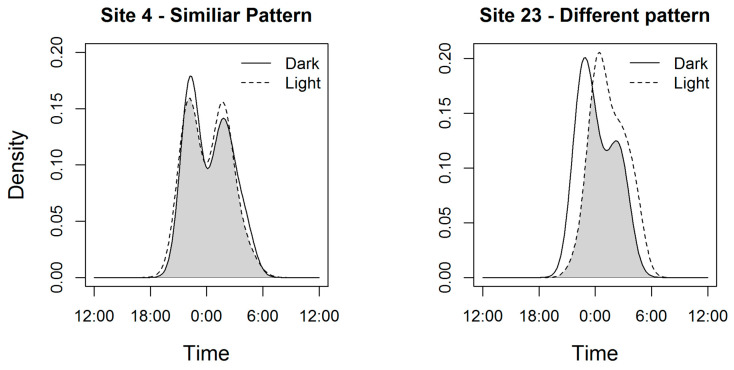
Examples of activity patterns at two different sites: hedgehog activity patterns did not change between treatments at site 4 (Δ = 0.93, *p* = 0.78), but did at site 23 (Δ = 0.70, *p* = 0.02), whereby hedgehogs shifted their activity to become active later at night when the light was on. The area shaded in grey is where the two patterns overlap (Δ).

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
