# Peer review of "Effects of Artificial Light at Night (ALAN) on European Hedgehog Activity at Supplementary Feeding Stations"

_animals, 2020, doi:10.3390/ani10050768_

Round 1

Reviewer 1 Report

This is a well-conducted study and a well-presented paper. I have only minor points to raise and hope that these can be addressed.

In the simple summary, the abbreviation ALAN is used the sentence before it is defined. Please swap the definition into the previous sentence.

Line 25: should read 'throughout the experiment, although some variations in individual hedgehogs were observed'

Line 34: italicise Erinaceus europaeus

Line 61: maybe would read better as '...high level of legislative protection..'

Line 66: should be clear that this is a rather unreliable estimate. Suggest change to  '. . . populations have reduced from a rough estimate of 1.5 million individuals in 1995 . . ."

Line 76: sing./plural error. Should read ' . . . the implications of this change for hedgehogs and their prey are relatively unknown.'

Line 219: 'an' should be 'and'

Comment: An interesting feature of the data that you do not comment on is the bimodality of the activity (feeding?) pattern shown in many cases. A tendency to bimodal feeding rhythms has been observed in some past field and captive studies (see p.163 of Reeve N.J. (1994) Hedgehogs, T & A D Poyser). The fact that this study has data that to some extent supports those observations is a small point of interest that could be mentioned.   

Reviewer 2 Report

The study describes an experiment in which a total of 33 hedgehog feeding sites in Great Britain were used to test whether hedgehogs change their activity and feeding behaviour when the area is illuminated or not.
All in all, this is a well thought out and methodologically sound study. The only criticism which I recommend to include in the discussion is the fact that the authors did their study at feeding places. As also the authors mention in the discussion, the attractiveness of easily accessible food often outweighs other risk considerations (like predators) and is therefore not to be equated with "normal" behaviour in places where no food is offering. In short, the attractiveness of the food may also have masked the normal reactions of hedgehogs (e.g. avoiding light).

Small mistakes:
- How exactly the "percentage of urban cover" was calculated? Which parameters were included (population density, building density)?
- Figure 1: Please explain what is meant by "number of blocks" (number of 10-minute intervals) and the unit (blocks per station per night). The figure should be understandable without reading the main text of the article
- It would be nice to know the duration of the experimental nights (min, max, mean) in order to know how many blocks would be possible at most/mean.
- References:
9) Write Harris in small letters
48) Space missing between "Behavior" and "of"
55) Year is missing
- Figure S1: It would be great if the station IDs (see TableS1 or FigureS2) are also included in this map!

Reviewer 3 Report

Nice little study about the potential impacts of ALAN on hedgehogs. I have one concern about the study and I wonder if that can be solved by revising the manuscript without going out in the field again. The authors compare 'dark' areas with 'light' areas but fail to state what 'dark' and what 'light' entails.... how dark were the dark areas? were there streetlights present in the area for instance? When a light was not provided by the authors but a patio light was used, were all patio lights of similar strength? i guess not... It appears as though the authors ignored this and did not measure the lumen at the sites, which is a large drawback and may have affected the results. 

Other comments 

Check grammar throughout. E.g. in the simple summary alone there are the following mistakes: L18: increased, L24: effects, L25: throughout experiment.. ? rephrase, L25: variations were or variation was, L28: and natural prey

Furthermore:

L46: it is stretching it a bit to say the hedgehog is threatened. LC on the iucn

L67: other countries... correct but you give two references from the Netherlands-. There is also evidence from e.g. Belgium and Sweden. Holsbeek et al. Hedgehog and other animal traffic victims in Belgium: Results of a countrywide survey. Lutra 1999, 42, 111–119. Krange. Change in the Occurrence of the West European Hedgehog (Erinaceus europaeus) in Western Sweden during 1950–2010; Karlstad University: Karlstad, Sweden, 2015.

L67-69: also see Hof et al (2019). Investigating the role of the eurasian badger (Meles meles) in the nationwide distribution of the western european hedgehog (Erinaceus europaeus) in England. Animals, 9(10), 759.

L87: what kind of food was provided. Same everywhere?

L91: was the other light at the site also recorded? i.e. what is dark? A nearby streetlight may have an impact for instance or other lights in the garden not directly aimed at the feeding station. There should have been a darkness measurement at each site. It is now not clear how large the difference between dark and light is and this is a serious deficit of the manuscript.

L104,105: how did you define the start of those blocks? What if at the end of a block a hedgehog was feeding and it was still feeding at the start of the next block? More detail is needed

L99 & L122-123: type of light used? Should have been always the same. Were the lumens recorded of patio lights? Are patio lights indeed comparable to the provided lights?

L131-132: what is sufficient data?

L132-134: can you rephrase/clarify? I don’t understand.

L184: de Molenaar should be De Molenaar. The same at the L322. It is a Dutch surname and they are as follows e.g. ‘Jan de Molenaar’ or Mr. De Molenaar

Can group size of hedgehogs be taken into account in the analyses?

Round 2

Reviewer 3 Report

Thank you for the response to my review. However, I noted that you did not add anything from the explanation you gave with regards to dark and light sites to the text. I suggest you do do this because more readers may have the same questions as I did. E.g. it is good to know that you asked if any features (such as a nearby streetlight) were likely to interfere with the experiment.

Author Response

We thank the reviewer for their comments. We have now added additional information into the methods as suggested by the reviewer. Please see Line 85-88 and Line 103-104.